# Anticancer Drug-Loaded Chitosan Nanoparticles for In Vitro Release, Promoting Antibacterial and Anticancer Activities

**DOI:** 10.3390/polym15193925

**Published:** 2023-09-28

**Authors:** Naushad Ahmad, Mohammad Rizwan Khan, Subramanian Palanisamy, Sonaimuthu Mohandoss

**Affiliations:** 1Department of Chemistry, College of Science, King Saud University, Riyadh 11451, Saudi Arabia; anaushad@ksu.edu.sa (N.A.); mrkhan@ksu.edu.sa (M.R.K.); 2East Coast Life Sciences Institute, Gangneung-Wonju National University, Gangneung 25457, Republic of Korea; spalanisamy33@gwnu.ac.kr; 3School of Chemical Engineering, Yeungnam University, Gyeongsan 38541, Republic of Korea

**Keywords:** chitosan nanoparticles, drugs, in vitro release, antibacterial, anticancer

## Abstract

Targeted drug delivery to tumor cells may be possible using nanoparticles containing human therapeutic drugs. The present study was carried out to develop cisplatin (CP) and 5-fluorouracil (FA) encapsulated chitosan nanoparticles (CSNPs), crosslinked with sodium tripolyphosphate (TPP) by an ionic gelation method and in vitro release, promoting antibacterial and anticancer activities. The prepared CSNPs, before and after CP and FA encapsulation, have been studied using various characterization techniques such as FTIR, XRD, SEM, and TEM-SAED patterning. The composites were well-dispersed, with an average particle size diameter of about 395.3 ± 14.3 nm, 126.7 ± 2.6 nm, and 82.5 ± 2.3 nm, respectively. In vitro release studies indicated a controlled and sustained release of CP and FA from the CSNPs, with the release amounts of 72.9 ± 3.6% and 94.8 ± 2.9%. The antimicrobial activity of the CSNPs-FA (91.37 ± 4.37% and 89.28 ± 3.19%) showed a significantly better effect against *E. coli* and *S. aureus* than that shown by the CSNPs-CP (63.41 ± 3.84% and 57.62 ± 4.28%). The HCT-116 cell lines were selected for in vitro cell cytotoxicity and live/dead assay to evaluate the preliminary anticancer efficacy of the CSNPs-CP and CSNPs-FA towards successfully inhibiting the growth of cancer cells.

## 1. Introduction

Over the last few decades, cancer has become the deadliest disease in the world [1]. It is one of the foremost causes of morbidity and mortality in the world. The challenge of treating tumors involves early-immune invasion, metastasis, and other biological characteristics [2]. Surgery, radiation, and chemical treatment are the standard therapeutic modalities [3]. The chemotherapeutic drugs 5-fluorouracil, cisplatin, gemcitabine, doxorubicin, and others are frequently utilized [4,5,6]. Although 5-fluorouracil has been used extensively to treat cancer, its half-life is only 10–15 min, and its in vivo retention duration is also limited [7]. Its clinical use was constrained by the requirement for frequent dosing. Therefore, the creation of a novel cancer treatment strategy is urgently required. New therapeutic platforms provided by nanotechnology may herald the arrival of the next generation of cancer therapies [8]. Since nanoparticles have been created to enhance the pharmacological and therapeutic benefits by reducing the harmful side effects of the drugs, their prospective application as drug carriers has been highlighted during the past few years [9].

Chitosan (CS) has recently received increased interest in the pharmaceutical and biomedical areas due to its beneficial biological characteristics, including biodegradability, biocompatibility, and nontoxicity [10]. The ratio of D-glucosamine to the sum of D-glucosamine and N-acetyl D-glucosamine can be used to assess the deacetylation degree of CS, which is useful information regarding the number of amino groups (-NH_2_) along the chains [11]. CS-based micro and nanoparticles can be created using a variety of techniques, including spray drying, ionotropic gelation, chemical crosslinking, precipitation, and emulsion [12]. Ionotropic gelation, a straightforward technique that has been extensively used in recent years, is the result of electrostatic interactions between protonated amino groups in the CS and sodium tripolyphosphate (TPP) anions [13]. TPP has gained popularity due to its nontoxicity, fast gelling ability, and electrostatic interaction with cationic CS. However, there are very few studies related to the antifungal activity of CS-based micro and nanoparticles and the incorporation of TPP into CS matrices [14].

A new generation of cancer treatment prospects has been opened up by the development of nanotechnology. In addition to extending the release time and the efficacy of the chemotherapeutic drugs, 5-fluorouracil (FA) and cisplatin (CP) can also be enhanced using CS nano-carriers [15,16,17]. CP is one of the most widely used anticancer agents. Among cancers, it has been found effective in the treatment of testicular, ovarian, bladder, cervical, head and neck, colorectal, and lung cancers [18]. In addition, CP is poorly soluble in water/oil phases, limiting the development of nanoparticles capable of carrying and encapsulating large amounts of drugs [19]. The administration of CP to tumors has been made safer through its encapsulation in a lipid–polymer system. Additionally, FA has been used to treat solid tumors [20]. It is most frequently used in clinical chemotherapy to treat rectum or colon cancers, as well as precancerous dermatoses. To increase the effectiveness of the FA drug, critical research is required to identify superior drug delivery mechanisms. Combining FA drug-loaded nanocarriers to create a combined nanoparticle delivery system for the delivery of FA to colon cancer cells has proven successful in treating colorectal cancer and preventing liver metastases [21]. Chitosan-based nanoparticles have shown tremendous potential as antibacterial agents. Researchers recently prepared CSNPs using ionic gelation by varying CS and TPP concentrations [22]. According to a similar study, the presence of TPP resulted in the potent inhibition of both E. coli and S. aureus, while CSNPs lacking TPP were less effective [23]. In addition, the development of CP and rituximab-loaded CSNPs, with linked surfaces as targeted cancer nano-formulations, shows the emerging potential of these nanoparticles in cancer therapy. Also, the sustained release of FA-loaded CSNPs, in vitro and in vivo, highlights the promising nature of CSNPs [15,16,17]. When compared to CP and FA solutions, CS had a prolonged release effect. Wang et al. reported CS nanolayered CP-loaded NPs for increased anticancer efficacy in cervical cancer [24], while Kim et al. reported an antitumor efficacy of CP-loaded NPs in tumor-bearing animals [25]. Additionally, Othayoth et al. described a vitamin adorned with CP-loaded NPs for chemoprevention and cancer fatigue [26]. CSNPs loaded with FA demonstrated sustained release effects in an in vitro release investigation, according to Reddy et al. [27]. Interestingly, Aydın et al. reported FA-encapsulated CSNPs for the evaluation of controlled release kinetics, and Sultan et al. reported the CP-loaded CSNPs as target-specific for combating cancer [28]. As a result, more research is needed to compare the sustained release, antibacterial, and anticancer effects of CSNPs as chemotherapeutic drug carriers.

In order to address the inherent drug carrier problems, chitosan nanoparticles (CSNPs) encapsulate two commonly used chemotherapy drugs, i.e., cisplatin (CP) and 5-fluorouracil (FA). Further, by using CSNPs as drug carriers, CP and FA can overcome their limitations, including poor solubility and frequent dosing. Both CP and FA are examined in this study for their sustained release effects. It is crucial to maintain therapeutic levels over time to achieve optimal therapeutic results. In contrast to previous research examining CSNPs as antibacterial agents, the present study explores the antibacterial effects of tripolyphosphate (TPP) incorporated into CS matrixes, which has been substantially underexplored. Additionally, CSNPs itself has shown antibacterial effects against both types of bacteria because of its amino moiety. Therefore, the synergistic effect of both CSNPs and drugs (CP and FA) against bacterial growth can be evaluated. A new approach for enhancing the effectiveness of cancer treatments is to combine CP and FA drugs within CSNPs. Furthermore, CSNPs delivery of CP and FA results in higher levels of cellular internalization, enhancing the cytotoxicity against cancer cells. It represents a novel approach to treating cancer by improving drug uptake and efficacy.

In the present study, we prepared TPP-crosslinked cisplatin (CP) and 5-fluorouracil (FA) encapsulated chitosan nanoparticles (CSNPs) via the ionic gelation method (Figure 1). The particle size, shape, structure, and drug release of CP and FA from the CSNPs were characterized. The physicochemical properties of the CSNPs, CSNPs-CP, and CSNPs-FA composites were investigated using various analysis methods, such as FT-IR, XRD, SEM, and TEM-SAED patterning, and in vitro drug release; the drug loading capability was also investigated. Moreover, the antimicrobial activity of the CSNPs, CSNPs-CP, and CSNPs-FA composites against bacteria such as Staphylococcus aureus and Escherichia coli exhibits good antibacterial properties. In addition, the anticancer effect of free CP/FA and CP/FA-loaded CSNPs was investigated in HCT-116 cells. This approach for the drug delivery of CP and FA-loaded CSNPs achieved a significantly higher cellular internalization and an enhanced cytotoxic effect compared to that of CP and FA alone.

## 2. Materials and Methods

### 2.1. Materials

Chitosan (50–190 kDa) and sodium tripolyphosphate were obtained from TCI Chemical, Seoul, South Korea. Cisplatin (MW; 301.1, purity, ≥99.9% trace metals basis) and 5-Fluorouracil (MW; 130.0, purity, ≥99%) were purchased from Sigma-Aldrich (St. Louis, MI, USA), and deionized water (DW) was used throughout the experiments. The human colon cancer cell line (HCT-116) was purchased from the Korean Cell Line Bank (Seoul, Republic of Korea).

### 2.2. Preparation of Drug-Loaded Chitosan Nanoparticles

Briefly, 1.0% (*w*/*v*) of chitosan (CS) was dissolved in a 0.5% (*v*/*v*) acetic acid solution. The pH was then lowered to 6.0 using sodium hydroxide (NaOH) [7,29]. The above solution was then added to 2.5 mL of 1.5 mg/mL sodium tripolyphosphate (TPP) under mechanically stirring at 600 rpm, and the reaction was maintained for 2.5 h at room temperature. Cisplatin (CP) and 5-Fluorouracil (FA) powder were precisely measured and mixed with DW to create a solution with 50 mg/mL. Dropwise additions of the prepared CP and FA solutions were introduced to the CS solution (CS:drugs, 1:1 ratio), which was then thoroughly mixed using probe sonication with a Q500 sonicator. Sonication was performed at 15 min intervals for 3 min at 40–70% amplification during the mixing procedure under 90 min of stirring. The CSNPs-CP and CSNPs-FA spontaneously generated under probing sonication after the addition of TPP. The CSNPs-CP and CSNPs-FA were isolated using ultracentrifugation at a g-force of 1232 for 15 min at 4 °C. Then, the supernatant was discarded, and the precipitate was freeze-dried and stored at 4 °C for use in subsequent studies. The same process was used to prepare CSNPs without the CP and FA solutions.

### 2.3. Fourier-Transform Infrared Spectroscopy (FTIR)

An FT-IR spectrophotometer was used to record the FTIR spectra of CS, TPP, CSNPs, CP, FA, CSNPs-CP, and CSNPs-FA on KBr pellets (Perkin-Elmer, Shelton, CT, USA). A total of 5 mg of the samples were exposed to the infrared spectrum at wavenumbers between 400 and 4000 cm^−1^, with a resolution of 4 cm^−1^ and 32 scans.

### 2.4. X-ray Powder Diffraction (XRD)

The structures and different crystallinity patterns of CS, TPP, CSNPs, CP, FA, CSNPs-CP, and CSNPs-FA were examined by X-ray diffraction using a device from the Shimadzu Corporation Kyoto, Japan. The diffraction patterns were collected at 25 °C over an angular range of 10 to 80°, with λ = 1.5406 radiation of CuKα at 40 kV and 30 mA.

### 2.5. Morphology

Field emission scanning electron microscopy was used to examine the morphologies of the bio-nanocomposite films (FESEM JSM-7600F, JEOL, Tokyo, Japan). Before viewing, one drop of a diluted solution of CSNPs, CSNPs-CP, and CSNPs-FA composites was released onto a carbon film. Tecnai F-12 JEOL-JEM 2100 transmission electron microscopy was utilized to analyze the NPs morphology. A drop of aqueously dispersed CSNPs, CSNPs-CP, and CSNPs-FA composites was released onto a copper grid that had been coated with carbon.

### 2.6. Size Measurements

The Zetasizer was used to measure the zeta potential, particle size, and polydispersity index (PDI) of the CSNPs, CSNPs-CP, and CSNPs-FA composites. Each nanoparticle sample, properly diluted with double-distilled water, was placed in the capillary cells before testing. The analysis was performed in triplicate at a temperature of 25 °C with a refractive index of 1.33.

### 2.7. Drug Entrapment

The encapsulation efficiency of the CSNPs-CP and CSNPs-FA composites was assessed using the ultrafiltration centrifugation method. In brief, 10 mg of the CSNPs-CP and CSNPs-FA composite powder samples were placed separately in 10 mL phosphate-buffered saline (pH 7.4), shaken for around 1.5 h using an orbital shaker, and centrifuged for 15 min at 4 °C with 1232 g-force. A total of 2 mL of supernatant was used for the absorbance, which was measured against a blank using the UV-Visible spectrophotometer (Optizen UV 3220, Daejeon, South Korea) at wavelengths of 380 nm for CP and 264 nm for FA. The EE% was obtained by deducting the free CP and FA drugs from the total amount added using the formula below:EE% = total amount of drugs − the amount of the free drugs/total amount of drugs × 100

### 2.8. In Vitro Kinetic Release

The assessment of the release percentages was conducted over a span of 48 h. In an in vitro release investigation, approximately 10 mg of each sample containing CSNPs-CP and CSNPs-FA composites was suspended in 10 mL of phosphate-buffered saline (with a pH of 7.4). These suspensions were subjected to continuous agitation at 37 °C, employing an orbital shaker set at 60 rpm. Subsequently, the samples underwent centrifugation, and the resultant supernatant was analyzed for the quantity of CP and FA using spectrophotometry at specific time intervals (0, 0.5, 1.0, 1.5, 2.0, 2.5, 3.0, 3.5, 5.0, 6.0, 9.0, 12, 18, 24, 30, 36, 42, and 48 h). To maintain sink conditions, any withdrawn solution was replaced with fresh PBS solution at predetermined intervals. Furthermore, in order to propose a release mechanism, the data derived from the in vitro drug release studies of the CSNPs-CP and CSNPs-FA composites were graphed and fitted to a first-order kinetic model. The selection of the best-fitting model was determined by calculating the correlation coefficient. Additionally, to gain insights into the release mechanism, the release data were subjected to fitting using the first-order kinetic model, with *F_max_* applied to CP and FA release profiles, in accordance with Equation (1).
F% = *F_max_* (1 − *e*^−*kt*^)(1)
where F% is the fraction of the accumulated drug released at time *t*, *F_max_* is the maximum amount of drug released, and *k* is the first-order release constant.

### 2.9. Antimicrobial Activity

The antibacterial assessments of the CSNPs, CP, FA, CSNPs-CP, and CSNPs-FA composites were conducted using overnight cultures of all bacterial strains cultivated in LB broth medium. The colony-forming unit method was employed for this purpose. In a nutshell, 50 μL of water-based solutions containing CSNPs, CP, FA, CSNPs-CP, and CSNPs-FA composites at a concentration of 50 μg/mL were added at various dilutions to 950 μL of microbial cultures developed for *S. aureus* (ATCC700376) and *E. coli* (ATCC 6538), both maintained at 37 °C. As a positive control, 50 μL of water was introduced into the cultures. Following 24 h of incubation at 37 °C in a shaking incubator, the bacterial culture solutions were obtained and subsequently diluted to achieve a concentration of 10^8^ CFU/mL. After the overnight incubation, the colony-forming units on each plate were enumerated, with each experiment being conducted in triplicate.

### 2.10. Cell Cytotoxicity

In this study, the cell cytotoxicity was investigated using an MTT (methylthiazolyldiphenyl-tetrazolium bromide) assay, which measures a cell’s ability to convert soluble MTT into insoluble formazan crystals. The cytotoxicity effect of CSNPs, CP, FA, CP-loaded CSNPs, and FA-loaded CSNPs was tested against the human colorectal carcinoma cell line (HCT-116). The cells were seeded at a density of 5 × 10^5^ cells/well in a 96-well plate and incubated under humidified conditions (5% CO_2_ and 37 °C) for 24 h. The experiment was repeated three times. After removing the cell-cultured supernatant from the wells, the cells were incubated for 24 h with different concentration of CSNPs, CP, FA, CP-CSNPs, and FA-CSNPs, ranging from 5 to 100 μg/mL. The samples were diluted in culture medium supplemented with 10% fetal bovine serum (FBS) and antibiotics. The control wells contained only cell culture medium. Over the incubation period, the medium was removed, and 20 μL of freshly prepared MTT solution (5 mg/mL) was added to each well. The cells were then incubated for an additional 4 h to permit the formation of MTT formazan crystals. Subsequently, the crystals were solubilized in 50 μL of DMSO, and the absorbance was measured at 570 nm. The cell viability was calculated using the following formula:cell viability (%) = absorbance of sample/absorbance of control × 100.

### 2.11. Live/Dead Cell Assay

To investigate apoptosis-associated changes in the cell nucleus, we employed the dual staining method using 4′,6-diamidino-2-phenylindole (DAPI) and propidium iodide (PI). HCT-116 cells (1 × 10^5^ cells/well) were seeded in a six-well plate and incubated for 24 h. Subsequently, the cells were treated with the highest concentration of the formulated CSNPs, CP, FA, CP-CSNPs, and FA-CSNPs (100 μg/mL) for 24 h. Following incubation, the cells were washed with 1 × PBS and fixed in a solution of methanol: acetic acid (3:1, *v*/*v*). After 10 min, the cells were stained with DAPI (1 mg/mL) and incubated for 20 min in the dark. Simultaneously, the cells were washed with 1 × PBS and stained with a 50 μg/mL concentration of PI for 10 min. Finally, the apoptotic associated morphological changes were observed using a Nikon Research Inverted Microscope ECLIPSE TS2R-C-AL (Tokyo, Japan).

### 2.12. Statistical Analysis

The data are shown as mean ± standard deviation after three times of each experiment. The significance level for the one-way analysis of variance used to determine the data statistical significance was set at *p* < 0.05.

## 3. Results

### 3.1. Surface Analysis of Drug-Loaded CSNPs

The Fourier transform infrared spectroscopy (FTIR) spectra of CSNPs, CP-loaded CSNPs, and FA-loaded CSNPs are shown in Figure 1. The functional groups in the samples of FTIR spectra reveal the presence and interaction of molecules that have been adsorbed, as well as coating molecules that have been bound to the surface layers of the CSNPs, CP-loaded CSNPs, and FA-loaded CSNPs. The FTIR spectrum of the major peaks of the pure molecules of CS, TPP, CP, and FA (Figure 1a,b,d,e) are defined from previously reported literature [30,31,32,33]. When compared with similar CSNPs, characteristic CS peaks were observed at 3328, 2879, 1591, 1419, 1063, and 709 cm^−1^ (Figure 1c) [34]. There are significant benefits to be gained from using CSNPs without encapsulation. A hypsochromic shift to 1641 and 1562 cm^−1^ in the FTIR spectra of CSNPs is the result of interactions between their two parent groups, NH_3_^+^ (CS) and phosphate (TPP). The corresponding major characteristic peaks of CP are different, i.e., shifting and disappearing from CP-loaded CSNPs at 3464 to 3422 cm^−1^ (Figure 1f). However, the narrow band remained nearly the same from 1639 to 1645 cm^−1^ due to the complexation of CP-loaded CSNPs present in the CP, which indicates the encapsulation of CP in the CSNPs–CP nano-formulation matrix. The peak for FA at 2989 cm^−1^ has shifted to 2898 cm^−1^ in the FA-loaded CSNPs (Figure 1g), which represents the CH_2_ stretching vibration of the polymer linkage chains. For FA, the peaks at 2831, 1719, 1432, 1239, and 736 cm^−1^ have shifted to 2791, 1699, 1419, 1241, and 778 cm^−1^, respectively, in the FA-loaded CSNPs, which represent C–H stretching, the amide C=O group, the NH_2_ groups, CO–NH_2_, and the C–O group of carboxylic acid, respectively.

The distinctive crystalline or amorphous structures of the nanoparticles are characterized using XRD measurements. Based on their unique diffraction peaks, the current study XRD analysis at 2θ identified the CSNPs, CP-loaded CSNPs, and FA-loaded CSNPs. The corresponding CS, TPP, CP, and FA pure molecule (Figure 2a,b,d,e) peaks are considered, as previously reported literature [35,36]. In contrast, the CSNPs exhibit peaks at both 2θ of 13.8° and 22.7°, indicating a significant degree of amorphous phase, as shown in Figure 2c. The peaks at 23.2°, 24.3°, 28.6°, 31.2°, 32.4°, 35.1°, 40.7°, 45.3°, 49.2°, 51.3°, 58.2°, 70.1°, and 74.5° represent the 2θ values for CP, while the peaks that disappeared at 13.8° and 22.7° represent the 2θ values for CSNPs in CP-loaded CSNPs (Figure 2f). In addition, the 2θ value shifted and disappeared from pure FA crystalline drugs, and the amorphous nature of the CSNPs, which is seen at 24.7°, 31.6°, 45.2°, and 56.7°, represents the 2θ values, indicating conformational changes, with highly amorphous characteristics, in CSNPs-FA due to FA attachment to the surface of the CSNPs (Figure 2g). Moreover, the disappearance of the FA peak indicates the entrapment of the FA drug inside the CSNPs, as well as the amorphous state of the encapsulated drug.

### 3.2. Morphology of Drug-Loaded CSNPs

The morphology of the CSNPs, CP, FA, CSNPs-CP, and CSNPs-FA was studied using scanning electron microscopy (SEM) analysis. The SEM images show that the CSNPs have a spherical morphology, and most of the particles are aggregated, with a diameter between 200 and 800 nm, as shown in Figure 3a,b, verifying the nano-sized, spherical shape of the particles, which are well dispersed [37]. CSNPs-CP and CSNPs-FA were separated from each other, with a size of 200–450 nm, which is due to the CP and FA drug encapsulation [38]. The strong interparticle interactions within this size of CSNP usually results in the formation of larger aggregates. However, as shown in Figure 3c–f, the CP-loaded CSNPs and FA-loaded CSNPs showed a rather good dispersity, which is crucial for optimal drug delivery [39]. Furthermore, there is a uniform surface roughness observed for CP and FA drugs in the CSNPs. The results suggest that the size and surface of CSNPs obtained in this study are in agreement with those of previous studies [40].

As demonstrated in Figure 4, the transmission electron microscopy (TEM) analysis properties of CSNPs, CSNPs loaded with CP, and CSNPs loaded with FA were diverse, spherical, and had smooth morphological features. The morphology of CSNPs indicated that they were smooth and spherical (Figure 4a). According to an earlier publication, the mean particle size of CSNPs created with TPP as a cross-linker was reduced from 292 to 127 nm [41]. The particle images of both CP-loaded CSNPs and FA-loaded CSNPs were found to be spherical, with a uniformly dispersed and smooth surface. CP-loaded CSNPs are between 50–150 nm in size, whereas FA-loaded CSNPs exhibit a smaller, uniformly dispersed spherical shape, measuring between 20–120 nm with (Figure 4b,c) [42]. As shown in Figure 4d–f, CSNPs, CP-loaded CSNPs, and FA-loaded CSNPs composites yielded average particle sizes of 395.3 ± 14.3, 126.7 ± 2.6 nm, and 82.5 ± 2.3 nm. The corresponding SAED patterns for the CSNPs, CP-loaded CSNPs, and FA-loaded CSNPs composites are shown in Figure 4g–i. The observed Debye–Scherrer rings are totally surrounded, representing the crystalline nature exhibited by the CSNPs, CP-loaded CSNPs, and FA-loaded CSNPs composites. Our optimized formulation, made up of CP and FA-loaded CSNPs, showed nanoscale range particles with a spherical surface according to TEM analysis, due to the penetration of surrounding water molecules.

### 3.3. Size Measurements of Drug-Loaded CSNPs

An alternative imaging method that can be used to study the particle size distribution is dynamic light scattering (DLS). A DLS nano-sizer, which measures the size distribution of nanoparticles, was used to measure the polydispersity index (PDI). Using the DLS, the hydrodynamic particle size of CSNPs, CP-loaded CSNPs, and FA-loaded CSNPs were found as peaks at 356.4 ± 12.7 nm, 138 ± 8.6 nm, and 106 ± 11.4 nm, respectively, with an excellent dispersity index of 0.48 ± 0.01, 0.32 ± 0.01, and 0.24 ± 0.02 [7]. Subsequently, the DLS analysis confirmed the nano-range of particles, and the smaller particle size is due to the CP and FA loading into the CS polymer chains. To assess the stability of the CSNPs, CP-loaded CSNPs, and FA-loaded CSNPs, zeta potential values were obtained. The zeta potential of CSNPs, CP-loaded CSNPs, and FA-loaded CSNPs was found to be −68 mV, −59.4 mV, and −44 mV. The zeta potential study, which indicated that the CSNPs exhibited a higher electrical conductivity in colloidal systems than did the CP-loaded CSNPs and FA-loaded CSNPs, was consistent with this change. According to earlier studies, stable nano-formulations have a zeta potential at −30 mV [43]. In addition to the size and shape of the nanoparticles, their charge, as well as their hydrophobicity and hydrophilicity, can affect target cells. In order to improve cell transfection, positively charged nanoparticles will interact with negative charged cells more favorably, thereby enhancing the effectiveness of the transfection process [44].

### 3.4. Encapsulation Efficiency

The nature of drug molecules and carrier materials, as well as the drug concentration, play an important role in drug encapsulation and loading into nanoparticles. The CP and FA carrying capacity, in terms of encapsulation efficiency, is calculated. CP and FA loaded CSNPs successfully, with an encapsulation efficiency of range at 66.43% and 69.69%, respectively. FA-loaded CSNPs have a higher encapsulation efficiency than do CP-loaded CSNPs. Previously, it was reported that CP and FA affect significant changes in the encapsulation efficiency of CSNPs by the elevation of CP and FA amounts during the formation of nanoparticles. The above results are in good correlation with those reported in a previous study [7].

### 3.5. In Vitro Drug Release from CSNPs

A time-dependent drug release profile was performed under physiological conditions while CP-loaded and FA-loaded CSNPs were submerged at 37 °C in PBS (pH 7.4) to assess their drug release profiles. Desorption, erosion, degradation, reabsorption, and diffusion mechanisms of the polymeric network resulted in the release of the CP and FA from the prepared CSNPs [45,46]. In order to be efficient, a drug delivery system should have a high degree of drug association. For example, a typical biphasic release trend was observed in CP-loaded CSNPs where approximately 66.4 ± 1.6% of CSNPs were released within the first 9 h of the study, while the remaining 72.9 ± 3.6% of the drug was released in 48 h [21]. In addition, an initial burst in the release of FA-loaded CSNPs was observed within 9 h, and 84.2 ± 2.1% of FA drug was released, followed by a sustained release profile for 48 h at 94.8 ± 2.9%, indicating the potential of CSNPs-FA as a sustained drug delivery system [24]. The hydrophobic nature of the drugs allowed for a high EE, as they are able to enter nanocarrier systems more easily. These results indicate an ionic interaction between the CSNPs and TPP for CP and FA. According to the work of Zaman et al. [47], the drug was released in a burst of about 35% over a period of 24 h, while in our investigation with CP- and FA-loaded CSNPs, about 66.4 ± 1.6% and 84.2 ± 2.1%, respectively, of the drug was released in the first 9 h. Our results suggest that the higher amount of CP and FA drugs dispersed or near the surface of the CSNPs may account for the burst release [48]. In 48 h, CP and FA drugs were released from the CSNPs in amounts of 72, 9.3% and 94.8%, respectively. On the other hand, the obtained dissolution results were found to fit the first-order kinetic model well. This model suggests that the CP and FA kinetic release constant (k) did not change throughout the study. Despite CSNPs-FA exhibiting a ~22% higher maximum drug release amount (Fmax) than that shown by CSNPs-CP, their kinetic release constant seems to be 0.198 and 0.221 in CP-loaded CSNPs and FA-loaded CSNPs, respectively, as described in Table 1 and Figure 5. By partitioning between CSNPs and the surrounding aqueous phase, we suggest that both CP and FA-loaded CSNPs release profiles are controlled by diffusion mechanisms. From the graph, the sustained release rate can be attributed to the intermolecular interaction formed by the dispersion of the CP-loaded CSNPs and the FA-loaded CSNPs matrix [49].

### 3.6. Antimicrobial Activity of Drug-Loaded CSNPs

We prepared CSNPs with efficient antibacterial activity by loading CP and FA drugs in the CSNPs and validated the antibacterial effect of CSNPs, CP, FA, CP-loaded CSNPs, and FA-loaded CSNPs against Gram-negative and Gram-positive bacteria through colony-counting antibacterial experiments. For *S. aureus* and *E. coli* (Figure 6), the bacterial colony numbers of the FA-loaded CSNPs group were considerably greater than those of the CSNPs and CP-loaded CSNPs groups [50]. In contrast, the number of colonies substantially decreased after the addition of CP and FA drugs to the CSNPs. The antibacterial killing rates of CSNPs, CP, FA, CP-loaded CSNPs, and FA-loaded CSNPs against *E. coli* were 35.12 ± 6.30%, 36.02 ± 4.21%, 38.15 ± 5.74%, 63.41 ± 3.84%, and 91.37 ± 4.37%, respectively. The same trend was seen in the antibacterial killing rates of CSNPs, CP, FA, CP-loaded CSNPs, and FA-loaded CSNPs against *S. aureus*, which were 28.35 ± 4.78%, 27.81 ± 3.92%, 31.20 ± 4.63%, 57.62 ± 4.28%, and 89.28 ± 3.19%, respectively. This indicates that the addition of CP and FA drugs improves the antimicrobial properties of CSNPs. The antibacterial killing rate of FA-loaded CSNPs increased to 91.37 ± 4.37%, which was higher than that of the CP-loaded CSNPs and CSNPs; there was also a significant difference (*p* < 0.05), indicating that the combined effect of CP and FA drugs could enhance the antibacterial properties of the NPs [51]. It is important to consider the mechanism of action of the drug-loaded nanoparticle formulation when determining the antibacterial efficacy. It is evident that CP and FA act in a different manner in this case. The main purpose of CP is to treat cancer, whereas FA is primarily an antimicrobial agent that interacts differently with bacteria. In addition to the particle size and surface charge of CSNPs-FA and CSNPs-CP, they also possess stability factors that can influence their ability to interact with bacteria and provide drug delivery. These findings align closely with the previously discussed particle size measurements and drug release investigations.

### 3.7. Anticancer Properties

#### 3.7.1. Cell Cytotoxicity of Drug-Loaded CSNPs

In the present study, we delved into the potential anticancer properties of various formulations, including CSNPs, CP, FA, CP-loaded CSNPs, and FA-loaded CSNPs, when tested against HCT-116 cells via an MTT-based assay. The cells were treated independently with varying concentrations of these formulations, ranging from 5 to 100 μg/mL for a 24 h duration, as depicted in Figure 7. The control group, interestingly, displayed no signs of cytotoxicity. Conversely, as the cells were exposed to increasing concentrations of CSNPs, CP, FA, CP-loaded CSNPs, and FA-loaded CSNPs, cytotoxicity steadily escalated. To specifically elucidate the cytotoxicity outcomes, we offer the breakdown: CSNPs displayed cytotoxicity ranging from 4.6% to 21.5%, CP-loaded CSNPs exhibited cytotoxicity in the range of 37.7% to 68.9%, and FA-loaded CSNPs demonstrated cytotoxicity levels in the range of 50.7% to 83.7% against HCT-116 cells. It is worth noting that when the cells were treated with CP and FA individually, cytotoxicity was observed only at the highest concentration of 100 μg/mL, with CP recording 27.5% cytotoxicity and FA showing 38.1% cytotoxicity. However, the cytotoxic effect was more pronounced when utilizing CP-loaded and FA-loaded CSNPs. Among these results, it was evident that FA-loaded CSNPs exhibited the highest level of cytotoxic activity in comparison to the other formulations.

#### 3.7.2. Live/Dead Cell Assay of Drug-Loaded CSNPs

To further assess the anticancer potential of the developed CSNPs, as well as the individual pharmaceutical agents (CP and FA), along with the drug-loaded CSNPs (CSNPs-CP and CSNPs-FA), we employed live and dead cell assays, incorporating 4′,6-diamidino-2-phenylindole (DAPI) and propidium iodide (PI) fluorescence stains (Figure 8). DAPI serves as a commonly utilized agent to highlight nuclear alterations occurring during apoptosis and to quantify the percentage of apoptotic cells. This compound binds to cell nuclei and emits a blue fluorescence [52]. In contrast, PI is a membrane-impermeable dye employed to identify nonviable cells. During apoptosis, cells undergo increased permeability to PI, a molecule too large to penetrate live and viable cells. Consequently, PI staining offers insight into the extent of apoptosis within the cell population [53]. PI, a red fluorescent dye, exclusively attaches to necrotic and deceased cells, since it cannot infiltrate living cells. In the control group, the cells exhibited blue fluorescence due to DAPI staining. However, cells treated with CSNPs, CP, FA, CSNPs-CP, and CSNPs-FA displayed red fluorescence due to PI staining, indicating substantial cell death in the HCT-116 cells in comparison to those in the control group. It is noteworthy that the highest level of cell death was evident in cells subjected to drug-loaded CSNPs, particularly CSNPs-FA, as opposed to cells treated with CSNPs and individual drugs alone. These findings underscore that HCT-116 cells exposed to CP-loaded CSNPs and FA-loaded CSNPs exhibited a greater number of TUNEL signals compared to cells treated solely with free CSNPs, suggesting that drug-loaded CSNPs exhibit enhanced efficacy in the context of tumor therapy [25,35].

## 4. Conclusions

In conclusion, the ionic gelation method was used to prepare cisplatin (CP) and 5-fluorouracil (FA) encapsulated chitosan nanoparticles (CSNPs) crosslinked with sodium tripolyphosphate (TPP) to investigate the antimicrobial, cytotoxicity, and live/dead assay effects. The prepared CSNPs, CP-loaded CSNPs, and FA-loaded CSNPs composites were characterized using particle size, polydispersity index, zeta potential, FTIR, XRD, SEM, and TEM-SAED pattern. The as-synthesized CSNPs, CSNPs-CP, and CSNPs-FA were well-dispersed in aqueous media, and they formed a nanocomposite structure with an average particle size diameter of about 395.3 ± 14 nm, 126.7 ± 2.6 nm, and 82.5 ± 2.3 nm, respectively, which were confirmed by TEM images. In vitro release studies found that CP-loaded CSNPs and FA-loaded CSNPs composites can play a role in sustained drug release. In addition, the drug release kinetics followed the first-order model due to the higher value of the regression coefficient of the FA-loaded CSNPs (R^2^ = 0.9918) when compared to the CP-loaded CSNPs (R^2^ = 0.9840). The antimicrobial activity of the CSNPs-FA (91.37 ± 4.37% and 89.28 ± 3.19%) was significantly better than that of the CSNPs-CP (63.41 ± 3.84% and 57.62 ± 4.28%) against *Escherichia coli* and *Staphylococcus aureus*. Cytotoxicity and live/dead assay studies revealed that the anticancer action of the synthesized CP-loaded CSNPs and FA-loaded CSNPs composites was higher than that of the CSNPs and CP/FA drugs at the same time point. Therefore, CP and FA-loaded CSNPs have great potential for further clinical and biomedical applications.

## Data Availability

The data presented in this study are available upon request from the corresponding author.

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
