# Peer review of "Anticancer Drug-Loaded Chitosan Nanoparticles for In Vitro Release, Promoting Antibacterial and Anticancer Activities"

_polymers, 2023, doi:10.3390/polym15193925_

Round 1

Reviewer 1 Report

The manuscript focuses on the development of cisplatin and 5-fluorouracil encapsulated chitosan nanoparticles using the ionic gelation method. However, it appears that there is a lack of significant novelty, as similar work has been reported in the literature (e.g., doi:10.1038/s41598-021-04427-w; doi: 10.1016/j.ajps.2017.04.002; doi: 10.1016/j.jddst.2023.104371). The authors are encouraged to provide a comprehensive discussion highlighting the differences between this work and previously published similar works. Additionally, a separate section should be included that emphasizes the novelty of the study to distinguish it from others.

The antimicrobial activity of CSNPs-FA and CSNPs-CP is commendable, especially against E. coli and S. aureus. In contrast, their anticancer properties against MDA-MB-231 cells seem to be less effective. It is unclear if the Triple-negative Breast Cancer Cell Line MDA-MB 231 is insensitive to these drugs or if other factors come into play.

Comments and Questions:

1.      Relevance of the Cell Line: Why was the MDA-MB-231 cell line chosen for this study? As the reviewer, I feel this might not be the most suitable model to test the anticancer properties of CSNPs-FA and CSNPs-CP. Would the authors consider testing another cell line to provide a broader view of the nanoparticles' efficacy?

2.      Emphasis on Antimicrobial Activity: Given the prominent antimicrobial results, have the authors considered emphasizing the application of CSNPs-FA and CSNPs-CP in antimicrobial domains?

3.      Particle Size Discrepancy: There seems to be a discrepancy in the particle size mentioned in the abstract and the conclusions (395.3±14.3 nm vs 119.3±3.2 nm for CSNPs). Could the authors clarify this?

4.      Lack of Free Drug Controls: All the antimicrobial and anticancer experiments lack the controls of free FA and free CP. Without controls of free drugs, it is difficult for readers to ascertain whether the chitosan encapsulated FA and CP have improved efficacy compared to the free drugs.

5.      Release Mechanism: Can the authors elucidate the release mechanism of CP and FA from the CSNPs? Are there any factors that can potentially affect the drug release from the nanoparticles?

6.      Storage Stability: What is the storage stability of these nanoparticles? How long can they retain their drug-loading efficiency and antimicrobial and anticancer activities?

7.      Manuscript Structure: The current structure of the article seems to lack clarity in highlighting the primary focus of the research. Is it on the antimicrobial or anticancer properties of the nanoparticles? Restructuring the manuscript might provide clearer insights to readers about the study's objectives and findings.

In conclusion, while the study provides a comprehensive set of data, its novelty and focus need to be clarified and emphasized. The authors should consider revisiting their choice of cell line, potentially emphasizing the antimicrobial application, providing controls of free drugs, and giving more insights from their characterization studies.

The quality of the English language in the manuscript is overall good. However, the paper might require minor revisions for clarity, consistency, and smooth transitions.

Author Response

We thank Reviewer#1 for the favourable reception of our work and for highlighting the positive points in our study. We have revised our manuscript taking into great consideration all the comments and suggestions. Thank you for helping us to improve our manuscript. Please see the attachment. 

Reviewer 2 Report

Journal: Polymers (ISSN 2073-4360)

Manuscript ID: polymers-2610586

Title: Anticancer drugs-loaded chitosan nanoparticles for in vitro release, antibacterial and anticancer activities

 Int this study, the authors have prepared the cisplatin (CP) and 5-fluorouracil (FA) encapsulated chitosan nanoparticles (CSNPs) cross-linked with TPP by ionic gelatin method and intro release, antibacterial and anticancer activities. The synthesized products were analyzed by FTIR, XRD, SEM and TEM. The antibacterial studies were carried out against gram-negative ((E. coli) and gram-positive (S. aureus) bacteria while anticancer studies about MDA-MB-231.

This paper is recommended to accept for publication upon addressing the following points.

Introduction

·       Instruction is too long. Authors should consider to revise it.

·       Authors should provide the novelty of the work.

·       Why the authors have chosen the two drugs; CP and FA

·       Authors should add the scheme of the preparation of the used products. It is mentioned in the intro but it is missing.

·       What is the main difference of CSNPs after loading of CP and FA onto in FTIR figure 1. Author should consider to label the important bands related to loading materials.

·       From the XRD data, it looks like the CP and FA are crystalline, author should provide the SEM and TEM with corresponding SAED patterns in order to see their morphology and structure.

·       What is the purpose to do the SEM analysis? What are the big particles in Figure 3 (c,d) and the white small-sized particles in (e,f).

·       Authors should provide the SAED patterns of all the three products CSNOs, CP and FA loaded CSNPs in Figure 4.

·       Why the size of CSNPs (~ 400 nm) is larger than that measured with loaded products (126 and 85 nm).

·       The authors have mentioned that they have tested the Free drugs (CP and FA) for antibacterial and anticancer. Where the results of these experiments. They are not present in Figs. 6 and 7.

·       Why the antibacterial activity of CSNPs-FA is better than CSNPs-CP. Authors should explain the reason behind these results. What about the anticancer activities of these two materials when compare the results?

None

Author Response

We thank Reviewer#2 for the favourable reception of our work and for highlighting the positive points in our study. We have revised our manuscript taking into great consideration all the comments and suggestions. Thank you for helping us to improve our manuscript. Please see the attachment. 

Round 2

Reviewer 1 Report

The authors have addressed my concerns. The manuscript could be published.